# How to Prevent Hostile Behaviors and Emotional Exhaustion among Law Enforcement Professionals: The Negative Spiral of Role Conflict

**DOI:** 10.3390/ijerph20010863

**Published:** 2023-01-03

**Authors:** María Ángeles López-Cabarcos, Analía López-Carballeira, Carlos Ferro-Soto

**Affiliations:** 1Department of Business Administration, Santiago de Compostela University, 15705 Santiago de Compostela, Spain; 2Department of Business Administration, Vigo University, 36310 Vigo, Spain

**Keywords:** police professionals, law enforcement, laissez-faire leadership, role conflict, hostility, emotional exhaustion, self-efficacy, interactional justice, meaning of the work, family–work enrichment

## Abstract

The nature and characteristics of the current work environment of law enforcement professionals point out role-conflict situations as one of the main reasons leading to the occurrence of hostile behaviors and the worsening of employees’ well-being. Precisely, this research analyzes the mediating role of role conflict between laissez-faire leadership and hostility or police professionals’ emotional exhaustion. To mitigate the negative effects of role-conflict situations, the moderating role of certain personal resources such as self-efficacy, and organizational variables such as interactional justice, the meaning of the work and family–work enrichment is also analyzed. Structural equation modeling and multigroup analysis are used in a sample of 180 police professionals. The results show that role conflict fully and positively mediates the relationships between laissez-faire leadership and hostile behaviors or emotional exhaustion. Moreover, self-efficacy and interactional justice moderates the relationship between laissez-faire leadership and role conflict; the meaning of the work moderates the relationships between role conflict and hostile behaviors, and family–work enrichment moderates the relationship between role conflict and employees’ emotional exhaustion. The huge relevance of the work of law enforcement professionals and its implications for society justify this research, which aims to highlight the importance of avoiding role-conflict situations to improve labor welfare and prevent counterproductive and unhealthy behaviors.

## 1. Introduction

Law enforcement is a public police based on a military structure whose purpose is the protection, care, welfare and maintenance of law and order [1]. In particular, law enforcement organizations develop a wide range of duties and responsibilities to ensure that these organizational objectives are met [2]. The demanding and complex nature of the work of law enforcement professionals, along with the increasing pressure put on them by the current ever-changing and disruptive work environment, can lead these professionals to live with incompatibilities between the job expectations and the job requirements [3]. At the same time, the organizational structure of the institutions where police professionals work can encourage the occurrence of the stressful situations and undesirable behaviors that deteriorate well-being at work [2]. In this way, the profession of law enforcement professionals can be considered stressful, and the occurrence of role-conflict situations is very common (Hofer, 2021) [3]. In fact, the nature of the work of police professionals and the characteristics of the institutions where they work can lead them to be involved in conflicting demands and requirements if the roles, obligations and responsibilities are not clearly defined, explained and specified [4]. So, law enforcement professionals are required to carry out their work under certain rules, standardizations and values that can provoke role-conflict situations, if the work expectations are not properly explained and understood. Role conflict can be defined as directions or expectations that are inconsistent with each other [5]. The prolonged exposure of police professionals to this kind of situation can lead them to exhibit counterproductive work behaviors such as hostility (Lawson et al., 2022) [6] and to feel emotionally exhausted (Hofer, 2021; Lambert et al., 2022) [3,7]. Examples of hostility behaviors are anger, disgust or resentment, mistrust, suspicion and cynicism, or repressed and indirectly expressed verbal and physical aggression [6,7].

The extremely complex work context of police professionals makes the decision-making style highly dependent on leadership style, a crucial variable to avoid negative work experiences [8,9]. The risky circumstances in which law enforcement professionals carry out their work justify the relevance of the leadership style to provide them the necessary support to guarantee effective interventions [8]. The most common leaderships within public work environments, such as that in which police professionals work, are passive styles such as laissez-faire leadership, in which the leaders do not participate in decision-making or assume the responsibilities that the organization requires of them [2,10]. In this sense, public work environments are characterized by the large number and relevance of protocols, norms and rules to be observed, which in some cases can leave people in the background. In this way, leaders, parapeted behind the rules and norms, avoid getting involved in the decision-making processes, discharging all the responsibility of complying with them, all of which can lead police professionals to show passive and avoidant behaviors [2]. Precisely, the lack of involvement of the leader in the decision-making processes and work duties can make it difficult for employees to meet job expectations that, according to challenge–hindrance stressor theory [11], can exacerbate stress situations arising from or resulting in the existence of role conflict. The reasons that justify these feelings and behaviors derive from the huge demands and challenges of the work of law enforcement professionals, together with the increasing requirements that arise from recent social, political or economic changes. While hostility means a negative feeling of antagonism, ill-will and denigration toward others [12], emotional exhaustion refers to the loss of resources caused by interpersonal demands [13]. Both hostility and emotional exhaustion have harmful consequences for the well-being of the organization and its members [14,15] and can be greatly aggravated by the role-conflict situations and the lack of the authority and involvement of the leaders [3]. This situation may set off a negative spiral in which police professionals experiencing high levels of role conflict can exhibit counterproductive and hostile behaviors toward colleagues or service users [14] and feel that their physical and psychological resources are completely exhausted [16,17].

To mitigate the presence of role-conflict situations in avoidance-based work environments, the conservation of resources theory [18] suggests that self-efficacy, which refers to the individual’s coping skills in the face of a wide range of challenging and stressful situations [19], and interactional justice, defined as the quality of interpersonal treatment that employees perceive in the workplace [20], can allow law enforcement professionals to better meet their job expectations. In turn, the meaning of the work, which refers to the core values that employees identify with the job [21] (Kristensen et al., 2005), and family–work enrichment, which is defined as the degree to which experiences in one role (family) improve the quality of life in the other role (work) [22], can ensure police professionals’ well-being and avoid inappropriate and hostile behaviors when expectations and policies are inconsistent or poorly defined. The public and vocational nature of the profession of law enforcement professionals [2,23], their challenging and stressful work environment, and the strong linkage between their work and their personal life [24] justify the choice of the moderating variables of the study.

On a sample of law enforcement professionals and using structural equation modeling and multigroup analysis, the study aims to address two main research questions: (i) the mediating role of role conflict on both employees’ emotional exhaustion and hostile behaviors in work environments where passive leaderships are present, and (ii) the role of personal resources and organizational variables as buffers of role-conflict situations in demanding and stressful work contexts. Hence, this study greatly contributes to the advancement of the knowledge about role conflict and its influence on employees’ well-being in at least three ways. First, it provides in-depth insight about the mediating role of role conflict in the relationships between passive leaderships and employees’ hostility or emotional exhaustion; second, it helps us understand the moderating role of self-efficacy and interactional justice to mitigate the negative consequences of role-conflict situations when passive leaderships are present; and third, it helps us understand the moderating role of meaning of the work and family-work enrichment to mitigate hostile behaviors and employees’ emotional exhaustion when role-conflict situations are present. These relationships have been scarcely studied in the past, so much more research is needed around them, especially in the context of complex, demanding and high-risk professions.

The rest of the paper is structured as follows. The first section describes the conceptual framework and poses the study hypotheses; the second section explains the methodology used; the third section outlines the main findings; and the fourth one discusses them. Finally, the fifth section points out the practical implications and suggests future lines of research.

## 2. Literature Review

The life-threatening experiences and complex situations to which police professionals are exposed (threat, violence, injury or even death) requires them to make precise, discretionary and authoritative decisions, as well as quick and agile interventions [2,25,26]. In many cases, law enforcement professionals who work in public contexts develop a vocational profession related to tasks such as being involved in helping others, protecting and providing humanitarian care, or achieving the public interest [27]. In turn, the institutions where law enforcement works are characterized by being highly bureaucratized and formalized show strong values as hierarchy, strong discipline, authority, high sense of mission and loyalty to the institution [2].

For this reason, leadership style becomes especially valuable in the context of law enforcement work, since it traditionally gives a lot of importance to such variables as rank, hierarchy, control, strong culture or centralized decision-making processes [8]. This means that those who act as leaders have a great deal of influence on the performance of their subordinates, which underlines the need for them to act from principles based on positive leadership. In these work contexts, effective leadership behaviors are extremely important for at least two reasons. First, the leadership style influences significantly subordinates’ work experiences, affecting also their well-being and the quality of the service offered [8,9]; and second, the public nature and prevailing values of the law enforcement work context lead leadership styles such as laissez-faire leadership to be the most common [2,8] but not necessarily the best.

Laissez-faire leadership can be defined as the leaders’ avoidance of and inaction over the responsibilities or duties assigned to them [10]. The avoidance-based leaders show indifference, avoid making decisions, do not meet responsibilities and refuse to use the authority associated with their roles [10,28]. Thus, while this leadership style does not provide feedback or guidance to subordinates, it can thwart the achievement of an organization’s goals and the expectations of the employees [29], affecting negatively their performance and well-being [30]. The large number of existing protocols, norms and rules in the public law enforcement work context can lead laissez-faire leaders to transfer to police professionals the responsibility of deciding their duties and obligations, trusting them with the decision-making process [2]. Laissez-faire leadership is not necessarily destructive per se, but the extremely dynamic, challenging and complex police work environment requires decisive, timely and effective leader interventions [29,30]. In fact, clarity, sense of direction and timely responses can become leadership characteristics highly valued by law enforcement professionals to properly develop their tasks [31]. Conversely, passive forms of leadership such as laissez-faire leadership may have more damaging effects on police work than more active–destructive leaderships [30], since police professionals may see their needs to be unfulfilled and feel less recognized and motivated at work [32]. At the same time, the lack of authority and involvement, especially in the decision-making processes of the laissez-faire leaders, can result in poorly defined and communicated policies, procedures or systems [33]. This leads police professionals to live uncertain, ambiguous and potentially stressful situations that, due to the very nature of their profession, they are forced to resolve adequately. Thus, law enforcement professionals are required to assume multiple roles that can result in unclear objectives and expectations that are ignored by passive leaders, experiencing stressful situations that can worse their job stress levels [34] and well-being [17], while increasing their counterproductive work behaviors [14] and role-conflict situations [35]. Despite their relevance, the role conflict experienced by employees when laissez-faire leadership is present has not been studied in depth by the previous literature and, even less, whether the law enforcement work context is considered.

Role conflict can be defined as the incompatibility between the subject’s job expectations and the job requirements [5]. The discrepancies between employees’ expectations about their own roles at work and the roles to be performed arise when the tasks, the duties and the responsibilities to be done by the employees are not clearly defined, explained and specified [4]. So, employees can be seen involved in conflicting requirements, competing demands and situations where the lack/presence of adequate/inadequate resources can hinder the effective development of their duties and responsibilities [36,37]. Law enforcement professionals can also live situations of role conflict when their work expectations are misaligned with respect to their daily duties and responsibilities as police professionals [3,34]. In this sense, the public nature of police work and the special features of the organizations where they work are the most common sources of role-conflict situations. In other words, police professionals can deal with leaders who provide inconsistent guidelines, feel that they do not receive the support required by the institution to meet the job objectives and contribute to public service, and perceive that their performance evaluation is misaligned with their work [3]. Moreover, police professionals deliver services that citizens can consider negative or undesirable either because citizens do not approve of the actions or decisions taken or because they do not understand them. The result is that law enforcement professionals find it difficult that citizens engage with them or with their work [2,3]. In this complex work context, law enforcement professionals must undertake their work according to certain rules, standards, operational processes and values that many times do not fit with their own work expectations, which generates serious situations of role conflict. This is particularly serious in work environments characterized by uncertainty, complexity and the need to make decisions in a timely manner, as is the case for law enforcement professionals. As a result, the pressure that employees face by experiencing role-conflict situations can lead them to unfavorable and stressful work experiences with negative consequences for job involvement [4], job satisfaction [38], well-being [17] or role stress [34]. The stress and frustration created by role conflict can lead these professionals to not knowing what to do or how to do it in moments when the welfare or life of a citizen is at stake. The prolonged exposure of law enforcement professionals to this kind of situation can lead them to display unhealthy and counterproductive work behaviors, such as hostility or corruption, and to feel that their physical and psychological resources are completely exhausted [3,16,17].

Hostility refers to the negative evaluation of others’ behaviors that includes antagonism, ill-will and denigration [12]. Hostile employees show negative feelings such as irritation, resentment, negativism, sense of guilt and suspiciousness toward others (colleagues or third parties). These employees can be more prone to perceiving the behaviors of others as aggressive and threatening, can be willing to take revenge for past experiences and can even use violence in extreme and risky situations [12,39]. In this sense, hostility implies antisocial behaviors, cynical thoughts and feelings of anger, which, if prolonged over time, can lead employees to feel depressed and to use force against colleagues or third parties (in the case of law enforcement professionals, even against citizens). So, hostility is a counterproductive and deliberated work behavior that violates the organizational norms and values, and threatens the well-being of the organization and its members [14,15]. Work environments, such as those of law enforcement, characterized by their high complexity can result in highly stressful employees who ultimately can exhibit misconduct behaviors, especially dangerous ones. That is, hostile actions are especially serious when it comes to a public service that aims to safeguard and guarantee the predictability of the citizens’ lives.

Emotional exhaustion is defined as the loss of resources owing to interpersonal demands [13]. The exposure to excessive job demands and requirements in the work environment can lead law enforcement professionals to feel emotionally fatigued, without energy and without enthusiasm for developing their tasks [2,40]. Law enforcement work is especially stressful, which leads police professionals to feel their emotional resources are depleted [2]. Specifically, prolonged exposure to role-conflict situations aggravated by the lack of the authority and involvement of the leaders [3] can lead law enforcement professionals to develop their tasks in unhealthy work environments and feel emotional exhaustion. In addition, emotionally exhausted police professionals are, in turn, more likely to show harmful, hostile and deviant behaviors such as poorer interaction with citizens, use of force, substance use and anger or frustration, all of which have negative consequences for them, for organizations and for society [17]. Two important questions arise in this moment: what reasons can lead police professionals to change their attitudes from the vocation of service to hostility? Second: what reasons can justify that these professionals in their role as public servants end up emotionally exhausted at work? The in-depth analysis of the organizational variables that may be driving these changes and those that may moderate or eliminate them becomes mandatory.

### 2.1. The Mediating Role of Role Conflict between Laissez-Faire Leadership and Hostility or Emotional Exhaustion

The challenge–hindrance stressor framework states that there are two broad categories of job demands, namely challenges and hindrances. Although both are potentially resource-depleting demands, while hindrances thwart personal growth and goal achievement and cause harmful effects on employees’ well-being, challenges prompt learning, growth, work motivation and performance. According to this theoretical framework, laissez-faire leadership and role conflict mainly act as hindrances [41]. In turn, conservation of resources theory suggests that employees who lose their energy when they feel that their resources are being compromised or depleted become more selective and sensitive when using new resources. Avoidance-based leaderships and role-conflict situations can be stressful experiences for law enforcement professionals who may feel threatened by the risk of losing valuable resources to effectively carry out their job duties, while increasing the possibility of suffering emotional exhaustion and developing hostile behaviors toward others [3,16,17,35]. Therefore, the lack of involvement of laissez-faire leaders can result in situations when police professionals receive contradictory and inconsistent orders and instructions [4], which can lead them to feel confused because of the ambiguity and uncertainty about their roles and responsibilities, and not knowing what courses of action to follow [37]. These misaligned and conflicting role expectations can develop, in turn, an uncertain and stressful work environment where feelings of emotional exhaustion can arise [17] along with the tendency to exhibit hostile behaviors toward themselves, the institution and the entire society [16]. Thus, role conflict can be considered as a routine stressor of law enforcement professionals’ work with very negative consequences at the individual and organizational levels [3] that can be exacerbated by the presence of laissez-faire leadership. Although these arguments seem to be logical from the theoretical point of view, to date no research has analyzed the mediating role of role conflict in the relationships between laissez-faire leadership and employees’ hostility or emotional exhaustion. This objective is even more unknown if the law enforcement professionals’ work context is considered. Therefore, the following hypotheses are proposed. H1: role conflict mediates the relationship between laissez-faire leadership and employees’ hostility; H2: role conflict mediates the relationship between laissez-faire leadership and employees’ emotional exhaustion.

### 2.2. The Moderating Role of Self-Efficacy and Interactional Justice

Drawing on the conservation of resources theory, personal resources such as self-efficacy and organizational variables such as interactional justice can be effective tools to cushion the presence of role-conflict situations and can be also a source of motivation to improve employees’ well-being. The internal and proactive nature of self-efficacy [19] and the mainly public nature of the law enforcement work context justify the choice of self-efficacy and interactional justice as moderating variables in this study [2]. The objective is that police professionals become able to adapt to or protect themselves from the negative consequences of passive leadership, successfully shaping their work environments to avoid situations of role conflict.

Self-efficacy refers to individuals’ confidence about their capabilities to cope with stressful or challenging demands [19]. Self-efficacious employees positively evaluate their coping skills and trust their abilities to devote the necessary efforts to meet job requirements and overcome adversities in the work environment [42]. In addition, they feel driven to find challenging goals, showing motivation, energy and persistence in achieving them. In this way, self-efficacy can enable employees to cope more effectively with negative and unforeseen circumstances, improving their well-being [41]. Self-efficacy is a valuable personal psychological resource [41] that can help law enforcement professional manage and implement the courses of action required to control and shape the work environment successfully [43], especially when avoidance-based leadership is present, as in the case of law enforcement professionals. In other words, self-efficacy can allow law enforcement professionals to be able to fulfill their duties and assume their responsibilities correctly, achieving better job performance and well-being by cushioning the negative influence of laissez-faire leadership [41]. Precisely, police professionals can draw on their self-efficacy to reduce the insecurity, loss of control, uncertainty and ambiguity of highly complex work experiences, avoiding situations of role conflict derived from the leader’s passivity in decision-making and in the assumption of job duties. To date, no research has analyzed the moderating role of self-efficacy on the relationship between laissez-faire leadership and role conflict; therefore, the following hypothesis is proposed. H3: self-efficacy moderates the relationship between laissez-faire leadership and role conflict, such that the moderation effect will lead to weakening the positive relationship between the two variables.

Interactional justice is defined as the quality of interpersonal treatment that employees perceive in the workplace [20]. It refers to politeness, honesty and respect during the interpersonal communication process, thus emphasizing the human side of organizational practices [2,44]. Interactional justice also focuses on providing relevant information about policies, procedures or decision-making [16,20]. Neutral, transparent and consistent decision-making, along with polite, dignified and respectful interpersonal interactions, can provide employees high levels of motivation and involvement [45,46]. Likewise, the presence of interactional justice can help employees build effective work teams and better communication processes, all of which can significantly improve the quality of the service provided to citizens and the whole society [2,47]. Hence, respectful, courteous and fair interpersonal relationships can lead police professionals to feel more comfortable and confident, be more willing to take the initiative, and feel greater identification with the organization, as well as lead to greater adherence to rules and regulations [48] and encourage favorable behaviors intended to avoid situations of role conflict and misconduct [45]. It seems clear that positive interpersonal relationships, along with clear, truthful and consistent communication processes about the procedures used, can help law enforcement professionals align their work expectations with their daily duties and responsibilities, particularly when avoidance-based leadership is present. To date, no research has analyzed the moderating role of interactional justice on the relationship between laissez-faire leadership and role conflict; therefore, the following hypothesis is proposed. H4: interactional justice moderates the relationship between laissez-faire leadership and role conflict, such that the moderation effect will lead to weakening the positive relationship between the two variables.

### 2.3. The Moderating Role of Meaning of the Work and Family–Work Enrichment

Conservation of resources theory also suggests that the presence of certain job resources, such as the meaning of the work and family–work enrichment, can break the loss cycle in which employees with role conflicts are located by means of motivational processes focused on protecting them from the detrimental consequences of high job demands to achieve work-related goals while improving their well-being. The choice of the meaning of the work is justified by the mostly public and vocational nature of law enforcement work [23], and the family–work enrichment by the challenging, demanding and stressful work environment of law enforcement professionals and the strong connections between their work and their lives at home [24].

The meaning of work refers to the core values that employees identify with the job, such as the work content, the significance of the tasks performed and the job service contribution [21] (Kristensen et al., 2005). It involves knowing what employees do at work and the importance of what they do [49], considering also the social impact of their work and the possibilities for their personal growth [50]. In this way, perceiving that one’s work has a valuable meaning for others and for the organization, each employee can value it, which clearly improves his/her well-being, satisfaction and motivation [23]. Law enforcement professionals work in organizations whose main objective is to provide a valuable public service to society [23,51]. In this way, the meaning of the work can help law enforcement professionals feel that police work is significant and purposeful, and lead them to achieve their work goals, stimulating personal growth, learning and development [41]. A high sense of the meaning of the work can lead police professionals to rely more on their personal and job resources when they are experiencing role-conflict situations. At the same time, by feeling that their work has an important social role (a source of personal meaning and growth), employees may be less prone to engage in hostile behaviors [51]. To date, no research has analyzed the moderating role of the meaning of the work on the relationship between role conflict and hostile behaviors; therefore, the following hypothesis is proposed. H5: the meaning of the work moderates the relationship between role conflict and hostility, such that the moderation effect will lead to weakening the positive relationship between the two variables.

Family–work enrichment is defined as the degree to which experiences in one role (family) improve the quality of life in the other role (work) [22]. Employees with positive experiences in their family lives can transfer involuntarily those pleasant emotions to their work. In this way, the family domain can provide useful values that can be applied in the work domain, such as the skill to share, the flexibility to solve tasks, the way to cope with relational conflicts and the ability to commit to tasks and people [52]. The competencies and positive emotions developed in the family domain can be extremely relevant in police work, since police professionals are required to be psychologically and physically healthy to cope with highly stressful tasks while providing public value [24]. Furthermore, the transfer of resources from the family context can enable law enforcement professionals to improve their performance, satisfaction or supportive behaviors [53]. Hence, law enforcement professionals with high levels of family–work enrichment are less prone to be influenced by the depletion of the resources and, thus, to feel emotionally exhausted when role-conflict situations are present. To date, no research has analyzed the moderating role of family–work enrichment on the relationship between role conflict and employees’ emotional exhaustion; therefore, the following hypothesis is proposed. H6: family–work enrichment moderates the relationship between role conflict and employees’ emotional exhaustion, such that the moderation effect will lead to weakening the positive relationship between the two variables. Figure 1 shows the model proposed.

## 3. Methodology

### 3.1. Participants and Procedure

Data were obtained through a self-administered questionnaire to Spanish public police professionals from 1 May 2019 to 30 June 2019. The study population consisted of 1500 public police professionals. An explanatory cover letter and a questionnaire to inform about the purpose and justification of the study were distributed to a simple random sample of 200 police professionals. Two requests for collecting information in a completely confidential manner were placed to ensure a sufficient number of responses. In the end, 180 useable questionnaires were obtained (response rate: 90%; confidence level: 95% (*p* = q = *0.5*); sampling error: 7.30%). Of them, 88.4% of participants were men (*n* = 161) with a mean age of 46.29 (*SD* = 6.353), and 10.55% were female (*n* = 19) with a mean age of 41.736 (*SD* = 4.494). Their average service tenure was 24.016 (*SD* = 8.181) years and their workplace tenure was 13.733 (*SD* = 7.882) years. Structural equation modeling and multigroup analysis with the SPSS 22.0 AMOS package were used to analyze the data and the hypotheses proposed.

### 3.2. Instruments

The four-item MLQ-5X (short form) scale by Bass and Avolio [10] with a five-point Likert scale (1 means *never*; 5 means *always*) was used to measure laissez-faire leadership (LF). The scale for role conflict (RC) was taken from Kristensen et al. [21]. The scale included four items based on a five-point Likert scale (1 means *never* and 5 means *always*). It included items such as, “Are contradictory demands placed on you at work?” To measure hostility (HO), two items from the aggression questionnaire, refined version scale by Bryant and Smith [54] with a five-point Likert scale (1 means *never*; 5 means *always*) was used. It included items such as, “At times I feel I have gotten a raw deal out of life.” To measure emotional exhaustion (EE), the five-item MBI-GS scale by Maslach and Jackson [13] with a seven-point Likert scale (0 means *never*; 6 means *always*) was used. It included items such as, “I feel emotionally drained by my work.” To measure self-efficacy (SE), the three-item scale by Luthans et al. [55] with a six-point Likert scale (1 means *strongly disagree*; 6 means *strongly agree)* was used. It included items such as, “I feel confident in representing my work area in meetings with management.” To measure interactional justice (IJ), the four-item scale by Moliner et al. [56] with a five-point Likert scale (1 means *strongly disagree*; 5 means *strongly agree*) was used. It included items such as, “My immediate superior is very sincere with me.” To measure the meaning of the work (MW), the three-item ISTAS scale by Kristensen et al. [21] with a five-point Likert scale (1 means *never* and 5 means *always*) was used. It included items such as, “Is your work meaningful?” Finally, the foure-item SWING scale by Geurts et al. [22] with a fourpoint Likert scale (1 means *never*; 4 means *always)* was also used to measure positive family–work enrichment (PFW). It included items such as, “After spending time with your spouse/family/friends, you go to work in a good mood, positively affecting the atmosphere at work?”

## 4. Results

### 4.1. Common Method Bias

To avoid potential common method variance, data collection was controlled following the recommendations by Podsakoff et al. [57]. Respondents were requested to provide honest responses and the anonymity of their answers was ensured. The dependent variables of the study were placed after the independent ones, and tested and confirmed scales from previous studies were used. In addition, Harman’s single factor test [58] was used to model all the items as indicators of a single factor that represents method effects. The results revealed eight factors with eigenvalues above 1, which explained 73.605% of the total variance, and with the first factor explaining less than 28.25% of the total variance. To supplement the previous analysis and to assess the fit of the confirmatory factor analysis model [59], all the variables were loaded onto one factor. The results concluded that the single-factor model did not fit the data well and that the fit was significantly worse than that of the measurement model [*χ*^2^ (*df*) = 791.19 (398), *p* < 0.001, GFI = 0.768, RMSEA = 0.074, AGFI = 0.729, NFI = 0.788, TLI = 0.869, CFI = 0.881, CMIN = 1.989]. So, most of the variance in the data was explained by individual constructs, which allowed to confirm that common method variance was apparently not a significant problem in this study [57].

### 4.2. Model Analysis

Table 1 shows the means, standard deviations, simple correlations and estimated reliabilities of the variables used in this study. Goodness-of-fit of the measurement model showed good values: *χ*^2^ (*df*) = 550.30 (321), *p* < 0.001, GFI = 0.826, RMSEA = 0.063, AGFI = 0.780, NFI = 0.849, TLI = 0.917, CFI = 0.930, CMIN = 1.715.

Goodness-of-fit of the structural model also presented good values: *χ*^2^ (*df*) = 173.13 (84), *p* < 0.001, GFI = 0.885, RMSEA = 0.077, AGFI = 0.835, NFI = 0.904, TLI 0 = 0.934, CFI = 0.947, CMIN = 2.060. The results point out that laissez-faire leadership accounts for 17.9% of role conflict, and laissez-faire leadership joined to role conflict accounts for 11.9% of employees’ hostility and 25.7% of employees’ emotional exhaustion.

### 4.3. Mediation

Table 2 shows the results of the mediation effect of role conflict in the relationships between laissez-faire leadership and police professionals’ hostility, and between laissez-faire leadership and police professionals’ emotional exhaustion. The results have concluded that role conflict fully mediates the relationship between laissez-faire leadership and employees’ hostility. Two additional models were tested to fully confirm this result [60]. Table 2 shows the model-fit statistics and the path coefficients of the three models (partial mediation, full mediation and direct effects), confirming full mediation. The chi-square of Model 2 (total mediation) is higher than the chi-square of Model 1 (partial mediation) but not significantly different (Δ*χ*^2^ = 1.421, Δ*df* = 1); it is lower than the chi-square of Model 3 (direct effect) and is significantly different (Δ*χ*^2^ = 30.52, Δ*df* = 1). Sobel [61] and Goodman [62] tests also supported the mediating effect of role conflict (*Z* = 2.5658, *p* < 0.0102; *Z* = 2.6011, *p* < 0.0092, respectively). Table 3 shows the results of the bootstrap percentile confidence intervals method for direct and indirect effects. All previous results concluded a positive full mediation of role conflict in the relationship between laissez-faire leadership and employees’ hostility, supporting H1.

The results also concluded that role conflict fully mediates the relationship between laissez-faire leadership and employees’ emotional exhaustion. Table 2 shows the model-fit statistics and the path coefficients of the three models, confirming the fully mediation. The chi-square of Model 2 (total mediation) is higher than the chi-square of Model 1 (partial mediation) but not significantly different (Δ*χ*^2^ = 2.974, Δ*df* = 1); it is lower than the chi-square of Model 3 (direct effect) and significantly different (Δ*χ*^2^ = 43.11, Δ*df* = 1). Sobel [61] and Goodman (1960) tests also supported the mediating effect of role conflict (*Z* = 3.9514, *p* < 0.00007; *Z* = 3.9832, *p* < 0.00006, respectively). Table 3 also shows the results of the bootstrap percentile confidence intervals method for direct and indirect effects. All previous results also concluded a positive full mediation of role conflict in the relationship between laissez-faire leadership and employees’ emotional exhaustion, supporting H2.

### 4.4. Moderation

This paper also analyzed the moderating effects of self-efficacy and interactional justice in the relationships between laissez-faire leadership and role conflict, the meaning of the work in the relationship between role conflict and employees’ hostility and the positive family–work enrichment in the relationship between role conflict and employees’ emotional exhaustion. Multigroup analyses were used to test all the moderation effects. First, factor loading invariance among the groups was conducted by testing the significance of the chi-square differences between two CFA models, one in which the factor loadings were constrained so that they were the same in both groups, and the other without constraints. Regarding the moderating role of self-efficacy in the relationship between laissez-faire leadership and role conflict, Table 4 shows that the chi-square difference was significant (Δ*χ*^2^ = 12.219, Δ*df* = 5, *p* < 0.05), suggesting there was no factor loading invariance. Series of multiple group analyses were performed to analyze path differences. The results concluded that self-efficacy shows a factor loading variant in the relationship between laissez-faire leadership and role conflict, concluding its moderating role and supporting H3.

The moderating role of interactional justice in the same relationship was also performed. Table 4 shows a significant difference in the chi-square (Δ*χ*^2^ = 17.970, Δ*df* = 5, *p* < 0.01) and suggests there is no factor loading invariance, concluding the moderating role of interactional justice, supporting H4.

The moderating role of the meaning of the work in the relationship between role conflict and employees’ hostility was also performed. Table 4 shows that the chi-square difference is significant (Δ*χ*^2^ = 16.346, Δ*df* = 5, *p* < 0.01) and does not suggest factor loading invariance, concluding the moderating role of the meaning of the work, supporting H5.

Finally, the moderating role of positive family-work enrichment in the relationship between role-conflict and employees’ emotional exhaustion was also performed. Table 4 shows a significant difference in the chi-square (Δ*χ*^2^ = 11.938, Δ*df* = 5, *p* < 0.05), and suggests there is no factor loading invariance, concluding the moderating role of positive family–work enrichment, supporting H6. Table 5 show the paths and R^2^ coefficients in the moderation relationships of self-efficacy, interactional justice, the meaning of the work and positive family–work enrichment. All the above-mentioned results are shown in Figure 2.

The above results show that the relationships between laissez-faire leadership and role conflict, role conflict and hostility, and role conflict and emotional exhaustion differ according to the level of the moderating variables, but it is not clear how exactly they differ. The interaction terms are positive, suggesting that self-efficacy, interactional justice, the meaning of the work and positive family–work enrichment can weaken these relationships. Since the nature and the precise size of these effects cannot be estimated by merely examining the coefficients, the effects have been plotted to interpret them visually (Dawson, 2014). For the first moderating relationship (self-efficacy moderating the laissez-faire leadership–role conflict relationship), one new grouping variable was created categorizing self-efficacy into three levels (low, moderate and high) to predict the relationship already mentioned at each level of the moderator variable. Three different regression groups were obtained for this relationship (Figure 3a). The results show that a moderated level of self-efficacy has a strong regression effect (R^2^ lineal = 0.196) on the laissez-faire leadership and role conflict relationship (correlation value = 0.4427) (R^2^ lineal self-efficacy low = 0.187 (correlation value = 0.4324); R^2^ lineal self-efficacy high = 0.061 (correlation value = 0.2469)). Hence, it is demonstrated that the relationship between laissez-faire leadership and role conflict is weakened more with moderate and low levels of self-efficacy than with high levels.

The moderating role of interactional justice was also analyzed for the laissez-faire leadership–role conflict relationship. Figure 3b shows that a moderated level of interactional justice has a strong regression effect (R^2^ lineal = 0.155) on this relationship (correlation value = 0.3937) [R^2^ lineal high = 0.122 (correlation value = 0.3492); R^2^ lineal low = 0.069 (correlation value = 0.2626)]. Hence, it is demonstrated that the relationship between laissez-faire leadership and role conflict is weakened more with moderate and high levels of interactional justice than with low levels.

Figure 3c shows that a moderated level of the meaning of the work has a strong regression effect (R^2^ lineal = 0.113) on the role conflict–employees’ hostility relationship (correlation value = 0.3361) [R^2^ lineal high = 0.104 (correlation value = 0.3224); R^2^ lineal low = 0.018 (correlation value = 0.1341)]. Hence, it is demonstrated that the relationship between role conflict and employees’ hostility is weakened more with moderate and high levels of the meaning of the work than with low levels.

Figure 3d shows that a high level of positive family–work enrichment has a strong regression effect (R^2^ lineal moderate = 0.262) between role conflict and employees’ emotional exhaustion (correlation value = 0.5118 [R^2^ lineal low = 0106 (correlation value = 0.3255); R^2^ lineal moderate = 0.067 (correlation value = 0.2588)]. Hence, it is demonstrated that the relationship between role-conflict and employees’ emotional exhaustion is weakened more with high and low levels of positive family–work enrichment than with moderate levels.

## 5. Discussion

Drawing on challenge–hindrance stressors and the conservation of resources theories, the study results suggest that role-conflict situations can lead law enforcement professionals to exhibit hostile behaviors and be emotionally exhausted when passive leadership is present. The findings point out that role conflict fully mediates the relationship between laissez-faire leadership and hostile behaviors, and also the relationship between laissez-faire leadership and emotional exhaustion. In work environments characterized by passive leadership, such as those where law enforcement professionals work, employees can receive contradictory and incoherent orders and instructions derived from the lack of involvement, authority and decision of the leaders, which results in police professionals being unable to meet the work expectations. If this situation continues over time, police professionals can enter a negative spiral in which increasingly high levels of role conflict will result in higher levels of emotional exhaustion and the possibility of exhibiting hostile behaviors against colleagues, third parties or the institution. These results are in line with those obtained by previous research related to the negative consequences of role-conflict situations in the workplace [16,17]. The fully mediating role of role conflict underlines the importance of avoiding this kind of situation and the reasons that can generate them, such as passive leadership, as the way, in turn, of avoiding negative consequences on employees’ health and well-being and the quality of the service offered [14,34,35]. This must be the case in any type of work context and much more in the case of law enforcement professionals, given the nature of their work and its enormous implications to the lives of all citizens. Personal psychological resources such as self-efficacy and organizational variables such as interactional justice are very important to help employees meet their expectations and cushion role conflict’s negative effects when passive leadership is present. In this sense, the results confirm that self-efficacy and interactional justice moderate the relationship between laissez-faire leadership and role conflict. According to its internal and proactive nature and its motivational capacity, self-efficacy can act as a protective factor for police professionals [19], helping them mobilize their resources to effectively handle insecurity, uncertainty, ambiguity and the loss of control derived from the leader’s passivity, which ends up generating role-conflict situations. This result is in line with previous research that has also highlighted the protective role of self-efficacy on employees’ well-being [19,43]. Regarding interactional justice, respectful interpersonal relationships, fluid and comprehensible communication processes and transparency in decision-making procedures can lead law enforcement professionals to wish to take the initiative, increase the identification with the organization or achieve greater commitment to rules, procedures and regulations. All this can help them align their work expectations with the daily duties and responsibilities, weakening the negative effects of laissez-faire leadership and avoiding the possibility of occurrence of role-conflict situations. Quality interpersonal relationships along with all the positive effects derived from them can be powerful tools to compensate for the lack of positive leadership and avoid situations of ambiguity, uncertainty, complexity and conflict when performing work tasks. Previous research has also supported the role of interactional justice as a key driver of employee performance and well-being [16,45,46]. In sum, the results conclude that both self-efficacy and interactional justice are valuable moderating variables capable of successfully shaping police professionals’ work environments to avoid role-conflict situations.

Organizational variables such as the meaning of the work and family–work enrichment can protect employees from the harmful effects of police work environments and their stressful and risky demands, while improving their well-being. In this sense, the findings suggest that the meaning of the work moderates the relationship between role conflict and employees’ hostility. According to previous research, the mostly public and vocational nature of police professionals’ work leads the meaning of the work to help these professionals value their work by considering it significant and purposeful, despite its complexity and the problems derived from working in rigid and hierarchical institutions [23,51]. Specifically, the meaning of the work allows police professionals to meet the police work expectations and provide a valuable service to society while avoiding engaging in hostile behaviors. Therefore, strengthening the meaning of law enforcement work, remembering its public nature and reinforcing its vocational nature, the unwanted hostile behaviors of police professionals derived from role-conflict situations not correctly resolved can be avoided. In the same line, results point out the moderating role of family–work enrichment between role conflict and police professionals’ emotional exhaustion. Law enforcement professionals are involved in challenging, demanding and risky tasks, so family–work enrichment can provide them the skills, experiences and competencies needed to deal with and manage the stressful and complex police work and to be less or not at all emotionally exhausted. So, in the same line as that of other previous research, transferring valuable resources and positive experiences from the family to the work environment can minimize the loss of resources derived from a highly stressful job in which situations of role conflict commonly occur, preventing emotional-exhaustion situations [24,52,53].

## 6. Practical Implications and Future Research

Law enforcement is considered a stressful profession in which job demand hindrances such as role conflict are particularly present. So, given the huge importance of law enforcement professionals’ work and its implications for society, effective formulas are required to avoid the destructive consequences of role-conflict situations. This study provides further evidence that laissez-faire leadership can worsen role-conflict situations, leading police professionals to experience negative situations with destructive consequences on their well-being and the development of counterproductive behaviors. Thus, promoting constructive and positive leadership can thwart the presence of role conflict in law enforcement institutions. This type of leadership sets clear expectations, provides timely feedback to employees and encourages them to focus on their job performances to be able to fulfill their daily duties and responsibilities and avoid misalignment with their goals. Since leaders are especially responsible for preventing role-conflict experiences, it is important to design human development training programs through which they can acquire new abilities and competencies to provide support to law enforcement professionals, build fluid communication processes and enjoy greater responsibility in decision-making. Moreover, strengthening the potential of self-efficacy and interactional justice through socialization practices such as the promotion of public service values, the creation of communication channels to perceive the real impact of police work on citizens’ lives and the design of effective strategies based on codes of good practices can protect police professionals from the negative consequences of role-conflict situations. Likewise, psychological training to face dangerous interventions and the reinforcement of the vocational and public nature of law enforcement work can help law enforcement professionals strengthen the meaning and significance of their work while the possibility of occurrence of counterproductive behaviors is reduced. These programs can also be designed with the participation of citizens, whose opinions can be useful to ensure the creation and maintenance of the meaning of the work for police professionals. Finally, designing programs based on the promotion of synergies between the family and work contexts can motivate police professionals, providing them with the necessary skills to avoid emotional exhaustion and to cope with a very risky work environment. Public decision-makers and human resource managers should develop strategies and programs focused on promoting positive experiences at work, fostering healthy work environments. In sum, in the law enforcement work context it is essential to analyze in depth the destructive consequences of role conflict and passive leadership, while promoting healthy and happy work environments where employees can achieve better results at all levels.

## 7. Conclusions and Future Research Lines

The results of the study allow the presentation of the following main conclusions. In complex and demanding work environments such as the one under study, (i) positive and constructive leadership becomes crucial to guarantee both employees’ well-being and organizational performance, (ii) strategies to avoid role-conflict situations must be undertaken to correct and prevent their negative consequences on the work of employees, (iii) organizations and employees have job and personal resources to mitigate the negative effects derived from negative leadership and conflict situations, and (iv) public work contexts, mainly those where the vocation of the employees is present, must make big efforts to design strategies focused on the human dimension and labor welfare.

As with any empirical research, this study has some limitations. The data has been collected from a single source, a self-reporting measure; however, several procedures have been undertaken with the aim of ensuring that common-method variance bias is not a problem in this study. Although the variables considered in this study are especially relevant, future research could consider other hindrance demands (e.g., active–destructive leadership), other job resources (e.g., social support and public service motivation) or other organizational variables (e.g., extra-role behaviors) to delve into the influence of role conflict and its effects on other outcomes (e.g., job performance). In future research, it would also be of interest to analyze whether there are differences when considering private police institutions or other geographical areas, to highlight their idiosyncrasies and generalize the results obtained in this research.

## Figures and Tables

**Figure 1 ijerph-20-00863-f001:**
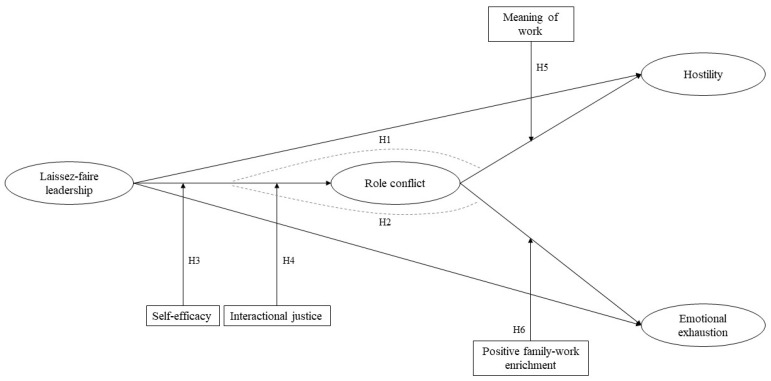
Proposed model.

**Figure 2 ijerph-20-00863-f002:**
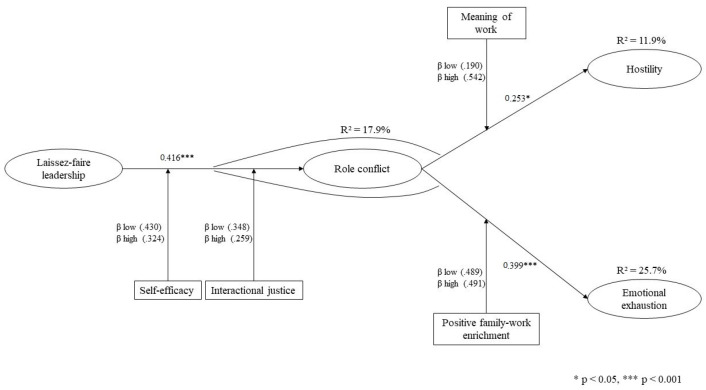
Structural model.

**Figure 3 ijerph-20-00863-f003:**
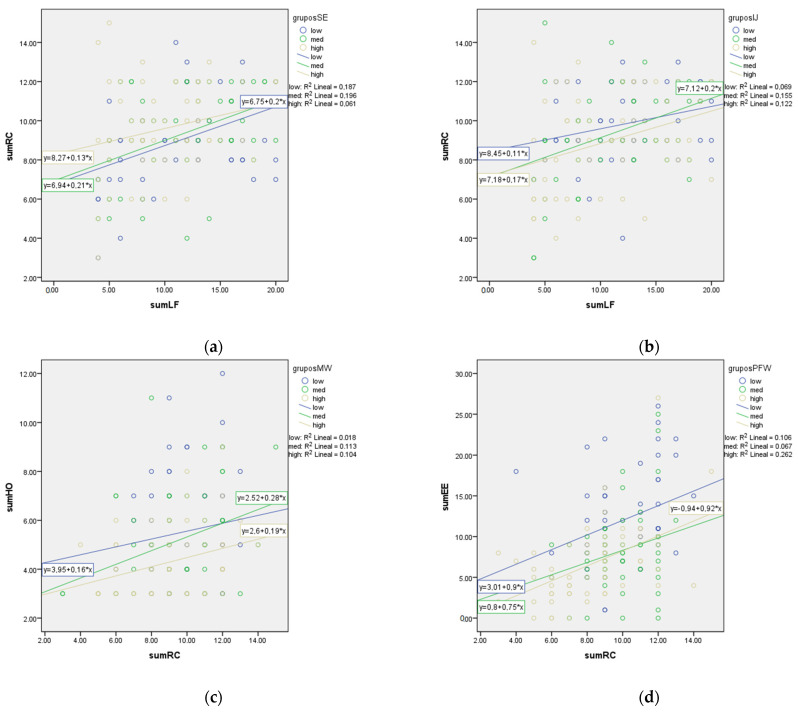
Graph of regression effects on different levels of self-efficacy, interactional justice, the meaning of the work and positive family–work enrichment. (**a**) Laissez-faire leadership vs. role conflict—self-efficacy moderator; (**b**) laissez-faire leadership vs. role conflict–interactional justice moderator; (**c**) role conflict vs. hostility—meaning of the work moderator; (**d**) role conflict vs. emotional exhaustion—positive family–work enrichment moderator.

**Table 1 ijerph-20-00863-t001:** Means, standard deviations, correlations and estimated reliabilities.

	M	SD	LD	RC	HOS	EE	SE	IJ	MW	PFW
LD	2.8361	1.1724	0.931							
RD	3.1278	0.74766	0.356 **	0.829						
HOS	1.6611	0.65395	0.176 *	0.282 **	0.703					
EE	1.7922	1.09102	0.267 **	0.426 **	0.453 **	0.906				
SE	3.9040	1.25073	−0.136	−0.216 **	−0.253 **	−0.377 **	0.861			
IJ	2.7722	1.09923	−0.726 **	−0.386 **	−0.170 *	−0.302 **	0.54 **	0.914		
MW	3.7259	0.78483	−0.341 **	−0.277 **	−0.310 **	−0.426 **	0.308 **	0.328 **	0.758	
PFW	2.4556	0.83223	−0.261 **	0.088	−0.018	0.039	0.068	0.172 *	0.359 **	0.824

Note: *n* = 180; * *p* < 0.05 ** *p* < 0.01; Cronbach’s α on the diagonal.

**Table 2 ijerph-20-00863-t002:** Fit results and path coefficients for structural equation models.

(a) Mediating Role of Role Conflict between Laissez-Faire Leadership and Hostility								
	*χ*^2^ (*df*)	GFI	RMSEA	AGFI	NFI	TLI	CFI	*χ*^2^/*df*
Model 1	53.764 (32)	0.944	0.062	0.904	0.947	0.969	0.978	1.680
Model 2	55.185 (33)	0.943	0.061	0.905	0.946	0.969	0.977	1.672
Model 3	85.705 (34)	0.917	0.092	0.866	0.916	0.930	0.947	2.521
Standardized Coefficients and (*t*-values)								
		Model 1			Model 2		Model 3	
RC←LF		0.413 (5.29)			0.417 (5.35) ***			
HO←RC		0.211 (1.96) *			0.212 (2.92) **			
HO←LF		0.128 (1.51)					0.211 (2.78) **	
(b) Mediating Role Of Role Conflict between Laissez-Faire Leadership and Emotional Exhaustion								
	*χ*^2^ (*df*)	GFI	RMSEA	AGFI	NFI	TLI	CFI	*χ*^2^/*df*
Model 1	106.618 (50)	0.968	0.08	0.856	0.933	0.951	0.963	2.132
Model 2	109.592 (51)	0.906	0.081	0.856	0.931	0.950	0.961	2.168
Model 3	153.702 (53)	0.875	0.0105	0.813	0.904	0.916	0.934	2.956
Standardized Coefficients and (*t*-values)								
		Model 1			Model 2		Model 3	
RC←LF		0.414 (5.31) ***			0.425 (5.38) ***			
EE←RC		3.76 (3.51) ***			0.460 (5.82) ***			
EE←LF		0.172 (1.72) *					0.325 (3.92) ***	

Note: Model 1: partial mediation; Model 2: fully mediation; Model 3: direct effects.* *p* < 0.05, ** *p* < 0.01, *** *p* < 0.001.

**Table 3 ijerph-20-00863-t003:** BC percentile method—direct and indirect effects.

Direct Effects	Effect	BootSE	*p*	BootLLCI	BootULCI
LF→RC	0.418	0.077	0.001	0.259	0.563
RC→HO	0.278	0.084	0.004	0.113	0.441
LF→RC	0.426	0.077	0.001	0.263	0.568
RC→EE	0.469	0.081	0.001	0.293	0.606
Indirect Effects	Effect	BootSE	*p*	BootLLCI	BootULCI
LF→RC→HO	0.116	0.044	0.003	0.043	0.216
LF→RC→EE	0.200	0.055	0.001	0.099	0.313

**Table 4 ijerph-20-00863-t004:** (a) Moderation Effect of SE (b) Moderation Effect of IJ (c) Moderation Effect of MW (d) Moderation Effect of PFW.

(a) Moderation Effect of SE								
Multiple Group CFA								
	*χ*^2^ (*df*)	*χ*^2^/*df*	Δ*χ*^2^ (Δ*df*)	RMSEA	CFI	*p*-value	Invariant	
Baseline (no constraints)	289.000 (172)	1.680		0.062	0.928			
Factor loading invariance	301.219 (177)	1.702	12.219 (5)	0.063	0.924	0.032	No	
Multiple Group SEM Models								Moderation
	*χ*^2^ (*df*)		Path invariance			*p*-value	Invariant	
Const LF→RC	294.326 (173)		292.810 (7)			<0.05	No	Yes
(b) Moderation Effect of IJ								
Multiple Group CFA								
	*χ*^2^ (*df*)	*χ*^2^/*df*	Δ*χ*^2^ (Δ*df*)	RMSEA	CFI	*p*-value	Invariant	
Baseline (no constraints)	267.704 (172)	1.556		0.056	0.937			
Factor loading invariance	285.674 (177)	1.614	17.970 (5)	0.059	0.928	0.003	No	
Multiple Group SEM Models								Moderation
	*χ*^2^ (*df*)		Path invariance			*p*-value	Invariant	
Const LF→RC	271.985 (173)		271.514 (7)			<0.05	No	Yes
(c) Moderation Effect of MW								
Multiple Group CFA								
	*χ*^2^ (*df*)	*χ*^2^/*df*	Δ*χ*^2^ (Δ*df*)	RMSEA	CFI	*p*-value	Invariant	
Baseline (no constraints)	271.889 (172)	1.581		0.057	0.939			
Factor loading invariance	288.235 (177)	1.628	16.346 (5)	0.059	0.932	0.006	No	
Multiple Group SEM Models								Moderation
	*χ*^2^ (*df*)		Path invariance			*p*-value	Invariant	
Const RC→HO	274.603 (173)		274.599 (7)			<0.10	No	Yes
(d) Moderation Effect of PFW								
Multiple Group CFA								
	*χ*^2^ (*df*)	*χ*^2^/*df*	Δ*χ*^2^ (Δ*df*)	RMSEA	CFI	*p*-value	Invariant	
Baseline (no constraints)	279.781 (172)	1.627		0.059	0.937			
Factor loading invariance	291.719 (177)	1.648	11.938 (5)	0.060	0.933	0.036	No	
Multiple Group SEM Models								Moderation
	*χ*^2^ (*df*)		Path invariance			*p*-value	Invariant	
Const RC→EE	283.601 (173)		283.591 (7)			0.05	No	Yes

Note: *p* < 0.1 (0.90 confidence), *p* < 0.05 (0.95 confidence), *p* < 0.01 (0.99 confidence).

**Table 5 ijerph-20-00863-t005:** (a) Paths and R^2^ Coefficients in Moderation Relationship of SE (b) Paths and R^2^ Coefficients in Moderation Relationship of IJ (c) Paths and R^2^ Coefficients in Moderation Relationship of MW (d) Paths and R^2^ Coefficients in Moderation Relationship of PFW.

(a) Paths and R^2^ Coefficients in Moderation Relationship of SE					
	Low	High	Low	High	
Relationships	β		R^2^		Mod. Confidence (%)
LF→RC	0.430	0.324	0.185	0.105	95
(b) Paths and R^2^ Coefficients in Moderation Relationship of IJ					
	Low	High	Low	High	
Relationships	β		R^2^		Mod. Confidence (%)
LF→RC	0.348	0.259	0.131	0.077	95
(c) Paths and R^2^ Coefficients in Moderation Relationship of MW					
	Low	High	Low	High	
Relationships	β		R^2^		Mod. Confidence (%)
RC→HO	0.190	0.542	0.041	0.312	90
(d) Paths and R^2^ Coefficients in Moderation Relationship of PFW					
	Low	High	Low	High	
Relationships	β		R^2^		Mod. Confidence (%)
RC→EE	0.489	0.491	0.258	0.253	95

## Data Availability

The data presented in this study are available on request from the corresponding author.

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
