# Peer review of "How to Prevent Hostile Behaviors and Emotional Exhaustion among Law Enforcement Professionals: The Negative Spiral of Role Conflict"

_ijerph, 2023, doi:10.3390/ijerph20010863_

Round 1
Reviewer 1 Report
First of All , Congratulations to Authors for taking up such a relevant topic which I personally consider it as a need of the hour .The authors presented systematic manner by using appropriate methodology.The way the authors executed the paper also appreciated
Few areas Authors can look in to
* Authors can elaborate Background information related to the topic in a detailed manner
* Little more clarity authors can bring methodology part
* Though the data collection happened before , Covid , as an reader and reviewer , it would be interesting if you can integrate Covid 19 as well to this research paper
* Minor language Correction is required .
Author Response
Reviewer 1
First of all, congratulations to authors for taking up such a relevant topic which I personally consider it as a need of the hour. The authors presented systematic manner by using appropriate methodology. The way the authors executed the paper also appreciated
Few areas authors can look in to:
* Authors can elaborate Background information related to the topic in a detailed manner
Following the reviewer suggestion, ‘Literature review’ section has been thoroughly revised to try to include arguments in a more detailed manner. Two new references have been added to do a more depth analysis of the arguments included in this section (p. 2, ln. 52).
* Little more clarity authors can bring methodology part
‘Methodology section has been reviewed in depth. This section includes three figures and four tables with detailed and complete information about the results obtained. All the analyses made have been explained in depth. Appropriate references have been used in this section to support all the analyses made. All this ensures that the reader can understand the methodology, the analyses and the results obtained.
* Though the data collection happened before, Covid, as a reader and reviewer, it would be interesting if you can integrate Covid 19 as well to this research paper
This study has used data collected between May 01, 2019 and June 30, 2019. So, these data refer to the period of time before Covid-19. For this reason, it is difficult to consider the consequences of this disruptive event in the research. It seems logical that extreme circumstances as those derived from the COVID-19 pandemic can modify the importance and influence of the variables involved in the study. Thus, Covid-19 or other disruptive and negative work environment events may aggravate and intensify the positive mediating effect of role conflict between passive leaderships and employees’ hostility behaviors or emotional exhaustion, and, at the same time, reinforce and strengthen the positive influence of moderating variables as self-efficacy, interactional justice, meaning of work and positive family-work enrichment. As the reviewer suggests, future research (research after Covid-19) will include, not the Covid-19, but the consequences of this disruptive event on variables related to human resources management.
* Minor language Correction is required.
The paper has been thoroughly checked in order to avoid grammar, syntax or structure/presentation flaws. To be sure that a formal/academic-specific style of presenting the ideas is used, a native English has revised the paper to help refine and improve the English in the paper.

Reviewer 2 Report
The current study by Lopez-Cabaracos et al., is an interesting examination of the relationship between key workplace measures including hostility, management style and the work-life balance. The layout of the article is well executed and the analyses are sufficient for the authors to make an argument in favor of their conclusion. I have a few suggestions in regards to some recommendations to help highlight/clarify their work.
- The inclusion of figures 3a-3d was a great idea. However, they are almost impossible to read. I’d advise you to clarify the axes labels for ease of the reader. I can see that you’ve mentioned this in the title, but it would help a lot to clarify this on the graph.
- I think Tables 4a through 5d are trying to present too much material. I think they could be scaled back accordingly.
- It’s clear that are trying to protect the anonymity of your participating agency. A worthy goal, but you don’t provide much detail, even at a general level, for us to better understand the representativeness of your sample presented here. How were they contacted? Was there compensation? What was the size of the agency? On page 8 you describe the sample, but it’s unclear what that means for the representation.
- This last point ties into the use of your language in the discussion section. While in the conclusion you do discuss some limitations, this can’t balance out the strength of some of your prior concluding statements. For example, you state, “The results of the study allow presenting the following main conclusions. In any work context…” Can you really say that? You have a single sample from one agency, and you never demonstrated that it’s a representative sample. I think you should tone this back for publication.
- Finally, I’d like you to consider your practical implications section on page 16. Reading through these carefully, do you think a modern police agency can be responsible for all of these different factors? You talk about promoting positive experiences at work, when we know that police officers are working dangerous jobs. Did you look to see what programs are already out there? Many agencies promote healthier lifestyles, talking to a therapist, etc. You should mention this briefly.
Minor notes; the acronym in your Table 1 doesn’t seem to mesh up with the text. Page 13, “..but it is no clear how exactly they differ
Author Response
Reviewer 2
The current study by Lopez-Cabarcos et al., is an interesting examination of the relationship between key workplace measures including hostility, management style and the work-life balance. The layout of the article is well executed and the analyses are sufficient for the authors to make an argument in favor of their conclusion. I have a few suggestions in regard to some recommendations to help highlight/clarify their work.
- The inclusion of figures 3a-3d was a great idea. However, they are almost impossible to read. I’d advise you to clarify the axes labels for ease of the reader. I can see that you’ve mentioned this in the title, but it would help a lot to clarify this on the graph.
It is better if Figures 3a, 3b, 3c, y 3d are shown together for comparison purposes. However, following the reviewer suggestion, we have enlarged the size of the figures to make them easier to understand for the reader.
- I think Tables 4a through 5d are trying to present too much material. I think they could be scaled back accordingly.
We have revised the information included in Tables from 4a to 5d. They include the necessary information to clarify all the analyses done. Moreover, we have put together all the tables related to the same type of analysis (4a-4b- 4c-4d; and 5a-5b-5c-5d) in order to precisely avoid including a higher number of tables in the paper.
- It’s clear that are trying to protect the anonymity of your participating agency. A worthy goal, but you don’t provide much detail, even at a general level, for us to better understand the representativeness of your sample presented here. How were they contacted? Was there compensation? What was the size of the agency? On page 8 you describe the sample, but it’s unclear what that means for the representation.
Following the reviewer suggestion, more information about the sample and the procedure used in the study was included in the manuscript (p. 8, ln 374-380).
- This last point ties into the use of your language in the discussion section. While in the conclusion you do discuss some limitations, this can’t balance out the strength of some of your prior concluding statements. For example, you state, “The results of the study allow presenting the following main conclusions. In any work context…” Can you really say that? You have a single sample from one agency, and you never demonstrated that it’s a representative sample. I think you should tone this back for publication.
We agree with the reviewer. The meaning of the idea was that the results can be extrapolated to ‘any work context whose characteristics are the same or similar to those of the work environment under study’… of course, not to any work context (in general). To clarify this for the reader, this sentence has been modified (p. 16, ln. 663-664).
- Finally, I’d like you to consider your practical implications section on page 16. Reading through these carefully, do you think a modern police agency can be responsible for all of these different factors? You talk about promoting positive experiences at work, when we know that police officers are working dangerous jobs. Did you look to see what programs are already out there? Many agencies promote healthier lifestyles, talking to a therapist, etc. You should mention this briefly.
We have analyzed many different police professionals’ work environments, not only in Spain but in other countries around the world. Most important is that we made in-depth interviews with several participants. Precisely, based on all this information we have built the ‘Practical implications’ section. They can sound ambitious but it is sure that they are necessary courses of action for avoiding the negative consequences of laissez-faire leaderships and role-conflict situations on the police professionals’ well-being.
Minor notes; the acronym in your Table 1 doesn’t seem to mesh up with the text. Page 13, “...but it is no clear how exactly they differ
The reviewer is true ante this error has been corrected.

Reviewer 3 Report
As attachement.

Author Response
Reviewer 3
Abstract
Although the abstract has listed the research findings, it should address the mediation effect is positive or negative
Following the reviewer suggestion, the sign of the mediating relationship has been included in ‘Abstract’ and ‘Results’ sections (p. 1, ln. 17; p. 10; ln. 455; ln. 466).
Introduction
- You mentioned in paragraph 1 on page 2 (lines 47-50) ‘The prolonged exposure of police professionals to this kind of situations can lead them to exhibit counterproductive and hostile work behaviors and feel emotionally exhausted (Hofer, 2021; Lambert et al., 2022; Lawson et al., 2022)’.
I think it is ambiguous and incomprehensible, please explain further. Although you have cited literature, please further explain why police will produce ‘counterproductive and hostile work behaviors’ and feel emotionally exhausted under long-term role conflicts. In addition, can you list the actual findings from existing research? For example, what are the types of counterproductive behavior and hostile behavior?
Paragraph 1 on page 2: Following the reviewer's suggestion, we have rewritten this part of the paper to be sure that it is clear and understandable for the reader (p. 2, ln. 48-52). Role-conflict is very common in all work contexts but especially in work contexts as that where law enforcement professionals work because off its demanding, risky and stressful nature, and the rigid, formalized and hierarchical structure of the police institutions. All this can lead police professionals to live conflicting demands and requirements if the roles, obligations, and responsibilities are not clearly defined, explained, and specified. In turn, prolonged exposure to role-conflict situations can trigger police professionals to show hostility behaviors, which is a type of counterproductive and deliberated work behaviors. Similarly, police professionals can feel that their physical and psychological energy is depleted, feeling emotional exhaustion.
Following the reviewer suggestion examples of hostile behaviors have been included in the manuscript (p. 2, ln. 50-52).
- You mentioned in paragraph 2 on page 2 (lines 56-62) ‘the most common leaderships within public work environments as that where police professionals work are passive styles such as laissez-faire leaderships, in which the leaders do not participate in decision-making or assume the responsibilities that the organization requires them (Bass & Avolio, 2004; Lopez-Cabarcos et al., 2022). Precisely, the lack of involvement of the leader in the decision-making processes and work duties can make it difficult for employees to meet job expectations that, according to challenge-hindrance stressor theory (Cavanaugh et al., 2020), can exacerbate stress situations arising from resulting in the existence of role-conflict’.
I couldn’t understand the style of police leadership mentioned in your manuscript. The police are a law enforcement agency that emphasizes discipline and obedience to orders. Therefore, both my research and the relevant literatures of the United States have showed that the police agency is a semi-military organization. The leadership of the police is usually authoritative. Based on this, I hope you can provide the cultural context of the police organization to which the research object belongs and explain the findings of the relevant research to present what you said is the police leadership style is ‘laissez-faire leaderships, in which the leaders do not participate in decision-making or assume the responsibilities that the organization requires them’.
Paragraph 2 on page 2: Following the reviewer suggestion and to clarify this issue, the sentence has been revised and rewritten (p. 2, ln. 61-66). A reference has been included to support the argument. “In this sense, public work environments are characterized by the large number and relevance of protocols, norms and rules to be observed, which in some cases can leave people in the background. In this way, leaders, parapeted behind the rules and norms, avoid getting involved in the decision-making processes, discharging all the responsibility in complying with them, all of which can lead police professionals to show passive and avoidant behaviors (Lopez-Cabarcos et al., 2022).”
Literature review
- It is recommended that you should explore the leadership models of the police agencies to which the subjects belong in your literature.
Before writing the paper, we have analyzed the leadership models of the police institutions not only in Spain, but in other countries. At least in Europe, we have found similar characteristics regarding the leadership models mainly when public contexts are considered.
- You mentioned in paragraph 1 on page 2 (lines 113-134) ‘In many cases, law enforcement professionals that work in public contexts develop a vocational profession related with tasks such as being involved in helping others, protecting, and providing humanitarian care, or achieving the public interest (Smith & Charles, 2010). In turn, the institutions where law enforcement works are characterized by being highly bureaucratized and formalized, and show strong values as hierarchy, strong discipline, authority, high sense of mission and loyalty to the institution (Lopez-Carbarcos et al., 2022). …second, the public nature and prevailing values of the law enforcement work context lead leadership styles as laissez-faire leaderships to be the most common (Fors Brandebo et al., 2019; Lopez-Cabarcos et al., 2022) but not necessarily the best’.
The above statement is not logical in its argument. For example, why do you say law enforcement agencies are characterized by high of bureaucracy and formalization and exhibit strong values… It was also stated that the communal nature and prevailing values of the law enforcement work environment led to leadership styles as laissez-faire leadership is most common. The above discussion is very abrupt, and you should do more sorting to be logical, rather than stringing together related literatures.
Paragraph 1 on page 2: This paragraph has been also revised to ensure the logical flow of the ideas regarding the role of leadership in the police professionals' work context (p. 3, ln. 120-137). The flow of ideas is: (1) vocational + (2) highly bureaucratized and formalized institutions = (3) leadership style becomes especially valuable (positive leaderships) + (4) two reasons to consider leadership style: influence subordinates’ work experiences, and not positive but negative (passive) leadership styles are the most common (not necessarily the best) when public work context are considered.
Law enforcement professionals are characterized by developing their work in public contexts, in which the use of standards, protocols and rules is very common. Thus, work environments, in which bureaucracy and formalization are more prevalent, can lead leaders to avoid participating in decision-making processes, ‘delegating’ this responsibility to subordinates. However, at the same time police professionals require decisive timely and effective leader interventions given the extremely dynamic, challenging, and complex police work. For this reason, effective and positive leaderships, but not negative-passive ones can be a powerful tool to help law enforcement professionals to properly develop their tasks.
- It is suggested that the relationship between the variables of hypothesis 1 and 2 can be specified as positive and negative.
Following the reviewers suggestion, it has been specified the positive mediation effect of role-conflict in the two mediating relationships proposed in the paper (‘Abstract’ and ‘Results’ sections) (p. 1, ln. 17; p. 10; ln. 455; ln. 466). In other parts of the papers before showing the results, it is specified only mediation or full mediation.
Methodology
- The method of sampling should be clarified and how?
Following the reviewer comment, we have included more explanation about the method of sampling in ‘Participants and procedure’ subsection (p. 8, ln 374-380).
- Why are these scales cited? What is their original reliability?
According to previous research, the scales of the study are the most widely used. If necessary, we can include a great number of papers using them previously. We have not changed one word of the original version of the scales. We have made all the necessary processes to ensure their appropriate use in this study. The Cronbach’s alphas of the scales used are: Laissez-faire, 0.931; Role-conflict, 0.829; Hostility, 0. 703; Emotional exhaustion, 0.906; Self-efficacy, 0.861; Interactional justice, 0.914; Meaning of work, 0.758; Positive family-work enrichment, 0.824.
Results
- It should provide references to prove whether each index value of ‘Godness-of-fit’ reaches the standard.
All the results included in the manuscript reach the recommended values but previous authors. This paper includes many different results and in most of the cases references of these authors supporting them have been included. The analyses made are very well-known by scientific community, so it could sound redundant to include their standards to reach.
Discussion
- This unit is well written. Especially, the ability to combine research findings with existing literature.
Thanks for the comments.
Conclusions
Paragraph 2 on page 17 (lines 663-637) should be placed in ‘future research’.
It is quite common to include conclusions and future research lines in the same section. Information about future research lines many times does not justify a separate section in the paper. According to the reviewer suggestion we have changed the name of the section ‘Conclusions and future research lines’.
